# Effects of Dietary Energy Profiles on Energy Metabolic Partition and Excreta in Songliao Black Pigs Under Different Ambient Temperature

**DOI:** 10.3390/ani14213061

**Published:** 2024-10-24

**Authors:** Kai Zhou, Dan Jiang, Xiaogang Yan, Guixin Qin, Dongsheng Che, Rui Han, Hailong Jiang

**Affiliations:** 1Ministry of Education Laboratory of Animal Production and Quality Security, Jilin Provincial Key Laboratory of Animal Nutrition and Feed Science, College of Animal Science and Technology, Jilin Agricultural University, Changchun 130118, China; 17767739581@163.com (K.Z.); jiangdan126@163.com (D.J.); qgx@jlau.edu.cn (G.Q.); chedongsheng@163.com (D.C.); 2Laboratory of Animal Nutrition Metabolism, Jilin Academy of Agricultural Sciences, Gongzhuling 136100, China; yanxiaogang1977@163.com

**Keywords:** ambient temperature, energy level, energy sources, energy metabolism, Songliao black pig

## Abstract

Energy profiles in feed are important because of the effects of differences in ambient temperature, feed energy levels, and energy sources on pig growth performance and energy metabolism. Therefore, we conducted a study to determine growth performance indicators, nutrient digestibility, and energy metabolism indicators of Songliao black fattening pigs and to test our hypotheses. At the same time, we used these indicators to give reasonable recommendations for optimizing feed energy profiles.

## 1. Introduction

The effect of different temperatures, feed energy levels, and energy structures on energy metabolism of pigs is not only an ancient problem, but also a recent problem. In recent years, many studies have proved that under different circumstances (genetic background, environmental temperature, life stage, etc.) [1], the demand for energy of pigs is not only a quantitative problem, but also a structural problem of energy carrier materials [2]. The previous energy nutrition system only considered the level of energy, but did not consider the problem of energy structure. Therefore, a comprehensive understanding of the demand and distribution of energy substances in growing and fattening pigs under different environmental temperatures, as well as the effects of feed energy levels and energy structure on pig growth performance, nutrient digestibility and energy metabolism, and the determination of optimal nutritional strategies, is essential for improving production efficiency and realizing precision nutrition.

Pigs are thermostatic animals with a suitable temperature range of about 17–24 °C [3]. If a suitable temperature environment cannot be provided, i.e., without increasing mineral energy to raise the ambient temperature in cold environments, pigs have to obtain more energy to maintain body temperature by increasing feed intake [4], and it has also been suggested that increasing the feed energy value can also mitigate the negative effects caused by low temperatures [5,6,7]. However, ‘energy value’, a single physical quantity, can only indicate the energy needs of animals and the energy nutritional value of feedstuffs, while ignoring the special needs of animals for the existence of different energy substances. He et al. [8] found that increasing the fat content of diets at low temperatures improved piglet growth performance. The study reported that when growing pigs in a low temperature, environment fat oxidation energy supply and heat production increase; an increase in the level of dietary fat can reduce the proportion of carbohydrate and protein oxidation heat supply and can improve the efficiency of nitrogen utilization [9,10,11,12,13]; the specific mechanism is not clear, but it shows that the energy needs of animals in a low-temperature environment are by no means the amount of problems, purely indiscriminately only to increase the total feed intake or the feed energy value; it will be accompanied by other energy carrier substances that are wasted and cause environmental stress [4].

Even at room temperature, the energy demand of pigs can never be measured only by using a single energy value (digestible, metabolizable, or net energy). Although the total energy value is the same, there are energy supply differences among the three major energy-supplying nutrients (carbohydrates, protein, and fat) in feed, and even though the energy substances can be transformed through certain metabolic pathways, the efficiency of energy supply and rate of utilization will be greatly reduced [14]. Therefore, to address such characteristics, the structural nature of energy supply still needs to be considered on the basis of total energy intake. Especially in the late growth of pigs, its demand for energy structure is more obvious [15,16], along with the efficiency of energy use, the characteristics of the use of different energy substances, excretion products, etc. There are great differences, such as the use of protein energy supply, that will lead to an increase in nitrogen excretion; the increase in the proportion of carbohydrates for energy supply will lead to an increase in the emission of CO_2_ [12], and in the fattening stage of the pig, an increase in the proportion of fat for energy supply may improve the efficiency of the use of energy, and reduce pollutant emissions at the same time.

There is a huge space for optimizing the technology of energy supply for different types of pig breeds and at different temperatures, and this technology is of great significance for the improvement in the economy and environment of animal husbandry. Based on this, this experiment took the Songliao black pig, a local pig breed in northern China, as the research object, and provided a theoretical basis for the optimization of the nutritional system of local pig breeds through the study of the precise supply of energy nutrition and the quantitative relationship between energy metabolism and the distribution of energy in the whole stage of growth and fattening under different ambient temperatures.

## 2. Materials and Methods

### 2.1. Animal and Experimental Handling

In this study, a 2 × 2 × 2 factorial array of treatments handled two temperatures (low-temperature, LT group: 10 °C; normal-temperature, NT group: 20 °C), two feed energy levels (normal-energy, NE group: 14.02 MJ/kg metabolic energy; high-energy, HE group: 15.14 MJ/kg metabolic energy), and two feed energy sources (LF group: low fat, HF group: high fat). The amount of soybean oil added in the HF group was 15% of the dietary digestive energy. The energy ratios of nitrogen-free efficiency regarding fat in LF and HF groups were 8.4:1 and 2.8:1. Thirty-two Songliao black finishing pigs with an initial body weight of 85.48 ± 2.31 kg at 170 days of age were randomly divided into 8 treatment groups, each with 4 replicates, and each replicate had 1 pig in the respiratory metabolism chamber for 6 days.

According to the Chinese Standard for Pig Feeding [17], four corn–soybean meal-based diets were designed to fulfill the nutritional needs of growing–finishing pigs weighing 75 kg [17]. The dietary formulations maintained uniform levels of digestible crude protein and amino acids, as detailed in the chemical composition analysis presented in Table 1.

Indirect calorimetry was carried out using eight open-circuit respiration chambers at the Animal Husbandry Branch of the Jilin Academy of Agricultural Sciences in Changchun, China, as detailed in a prior study [12]. The experiment was divided into four stages, with each period totaling 11 days. The pigs were fed at 8 a.m. and 4 p.m. every day. Each pig had a fixed feed intake every day and drank freely throughout the whole process.

### 2.2. Sample Collection

In this experiment, a digestion and metabolism test was conducted by the total feces collection method. Following a pre-feeding period of five days, fecal and urinary outputs from each pig were gathered over a subsequent six-day interval, and thoroughly combined, and then a 20% subsample of this mixture for each animal was preserved for an additional analysis. To stabilize nitrogen, the collected feces for each pig were placed in plastic bags and treated with a 10% hydrochloric acid solution. In total, 2% of the total urine was collected for 6 consecutive days and fixed with 10% sulfuric acid.

On the final day of the experiment, blood was drawn from the pigs into serum and EDTA tubes and centrifuged using a 5430 Eppendorf machine (Hamburg, Germany) at 3000× *g* for 10 min at 4 °C. The resulting plasma and serum were then allocated to storage tubes, immediately frozen in liquid nitrogen, and kept at −80 °C until needed for subsequent analyses.

### 2.3. Chemical Analysis and Calculation

In compliance with the guidelines of Chinese national standards, the content of the crude protein (CP) and ether extract (EE) in the diet, feces, and urine was accurately measured. Additionally, to ascertain the gross energy (GE) present in feed, feces, and urine, an isoperibol calorimeter (Parr 6300 Calorimeter, Moline, IL, USA) was utilized, employing benzoic acid as a calibration standard. These analyses were systematically replicated three times to ensure precision and reliability.

Throughout the six-day collection period, the average daily feed intake of each pig was recorded, along with their average daily weight gain and material weight ratio. These data allowed for the precise calculation of the GE intake, utilizing the GE values of their diets combined with their daily feed consumption statistics. For each pig, the intake of fecal energy (FE) and urinary energy (UE) was determined by averaging the daily excretion rates of feces and urine over a period of six days, combined with the established FE and UE values specific to each animal. Based on the measurements of oxygen consumption and carbon dioxide production, the calculations for heat production (HP), protein oxidation (OXP), carbohydrate oxidation (OXC), fat oxidation (OXF), and the respiratory quotient (RQ) were derived from an established equation relevant to these gas exchanges. The specific formula was consistent with previous studies [3]. Nutrient digestibility was calculated as the ratio between the nutrient digestion and intake. The formula for energy retention (RE) subtracts the GE, fecal energy (FE), urinary energy (UE), and heat production (HP) values. Additionally, the retention of energy as protein (REP) is determined by multiplying nitrogen retention (g/d) by 6.25 and then by 23.86 (kJ/g). The retention of energy as lipids (REL) is found by subtracting REP from RE. The urinary N or fecal N were calculated as the product of average daily urine output or dry matter for fecal discharge volume and N content in urine or feces.

### 2.4. Biochemical Marker Assays in Serum

Using commercial kits from Zhongsheng North Control Bioengineering Institute (Beijing, China), serum samples from pigs in the study were analyzed for various biochemical markers. These included albumin (ALB), high-density lipoprotein (HDL), low-density lipoprotein (LDL), total cholesterol (TCHO), triglycerides (TGs), total protein (TP), glucose (GLU), and blood urea nitrogen (BUN). Measurements were performed on an automatic biochemical analyzer (BS-400, Mindray, Shenzhen, China), adhering strictly to the protocols specified by the manufacturer.

### 2.5. Non-Target Metabolomics Profiling in Plasma

In the preparation of plasma samples, 100 μL aliquots were initially cooled to 4 °C. These samples were then treated with 400 μL of a chilled 1:1 methanol/acetonitrile solution to precipitate proteins. Following mixing, the samples underwent centrifugation at 14,000× *g* for 15 min at a temperature of 4 °C. Subsequently, the clear supernatant was extracted and concentrated using a vacuum centrifuge. Samples for an LC-MS analysis were reintroduced into the solution using 100 μL of an acetonitrile/water mix at a 1:1 volume ratio. The analysis was conducted using an ultra-high-performance liquid chromatography system (Agilent 1290 Infinity LC, Santa Clara, CA, USA), paired with a high-resolution mass spectrometer (AB Triple TOF 6600, Shanghai, China). This equipment facilitated metabolite analyses in both positive and negative ion detection modes. Using a Waters ACQUITY UPLC BEH column (1.7 µm, 2.1 mm × 100 mm), the study utilized a liquid chromatography–mass spectrometry (LC-MS) system to conduct analyses under both positive and negative ion conditions. Data acquisition in the study was managed through an information-dependent acquisition mode, utilizing Analyst TF 1.7.1 software from Sciex, based in Concord, ON, Canada. Subsequently, the LC-MS-generated data files were formatted into mzXML using ProteoWizard software (3.0.6428). Data processing was executed using the XCMS software (online 3.7.1), which handled tasks such as peak extraction, peak alignment, and retention time correction. The area of each peak was adjusted using the “SVR” method. Peaks detected in less than 50% of the samples within each group were excluded. Metabolite identities were established by querying both a custom laboratory database and an integrated public database. The identified metabolite data were then analyzed statistically.

### 2.6. Statistical Analysis

The statistical analysis and graphical representation of data were conducted using SPSS 25.0 (IBM-SPSS Inc., Chicago, IL, USA) and GraphPad Prism 8 (GraphPad Prism Inc., La Jolla, CA, USA), respectively. A one-way analysis of variance (ANOVA) facilitated the determination of statistical differences, supplemented by Duncan’s post hoc test for multiple comparisons. The Spearman rank test was employed for correlation analyses. Mean values ± SEM were reported, with statistical significance set at *p* < 0.05.

## 3. Results

### 3.1. Growth Performance

At 20 °C, the average daily gain was higher than 10 °C (*p* < 0.05), and the average daily gain in he and HF groups was higher than that in NE and LF groups (*p* < 0.05). When the temperature was 20 °C, the feed-to-gain ratio (F:G) was lower than 10 °C (*p* < 0.05), and F:G in he and HF groups was lower than that in NE and LF groups (*p* < 0.05) (Table 2).

### 3.2. Energy Metabolism

When the temperature was 10 °C, the total heat production, fecal energy, and urinary energy were higher than 20 °C (*p* < 0.05), and the deposition energy was lower than 20 °C (*p* < 0.05). There was no difference in total heat production, urine energy, and deposition energy between the HE group and NE group (*p* > 0.05), but fecal energy in the HE group was lower than that in the NE group (*p* < 0.05). There was no difference in total caloric production and urinary energy between the HF group and LF group (*p* > 0.05), but fecal energy and sedimentary energy in the HF group were higher than those in the LF group (*p* < 0.05). In terms of fecal energy, there was an interaction between the temperature and energy level and temperature and energy structure (*p* < 0.05). At 10 °C, fecal energy of the HE group was lower than that of the NE group (*p* < 0.05), while at 20 °C, fecal energy of the HE group and NE group had no difference (*p* > 0.05). In the NE group and he group, fecal energy at 10 °C was higher than that at 20 °C (*p* < 0.05). At 10 °C, fecal energy in the HF group was higher than that in the LF group (*p* < 0.05), while at 20 °C, fecal energy in the HF group was not different from that in the LF group (*p* > 0.05). In the LF group and HF group, fecal energy at 10 °C was higher than that at 20 °C (*p* < 0.05) (Table 3).

When the temperature was 20 °C, the energy of protein and fat deposition was higher than 10 °C (*p* < 0.05), OXPRO and OXFAT were lower than 10 °C (*p* < 0.05), and OXCHO had no difference compared with 10 °C (*p* > 0.05). There was no difference in protein energy, fat energy, OXCHO, and OXPRO between the he group and NE group (*p* > 0.05); OXFAT was higher than the NE group (*p* < 0.05). The energy of fat deposition, OXPRO, and OXFAT in the HF group were higher than those in the LF group (*p* < 0.05); OXCHO was lower than that in the LF group (*p* < 0.05); and there was no difference in energy of protein deposition between the HF group and LF group. In OXFAT, there was an interaction between the temperature and energy level (*p* < 0.05), and there was a strong interaction between temperature and energy structure (*p* < 0.01). At 10 °C, OXFAT in the HE group was higher than that in the NE group (*p* < 0.05), but at 20 °C, OXFAT in the HE group and the NE group had no difference (*p* > 0.05). In the NE group and he group, OXFAT at 10 °C was higher than that at 20 °C (*p* < 0.05). At 10 °C, OXFAT in the HF group was higher than that in the LF group (*p* < 0.05). At 20 °C, OXFAT in the HF group and LF group had no difference (*p* > 0.05). In the LF group and HF group, 10 °C OXFAT was higher than 20 °C (*p* < 0.05) (Table 4).

### 3.3. Digestibility of Nutrients

When the temperature was 10 °C, the energy digestibility and fat digestibility were lower than 20 °C (*p* < 0.05), but there was no difference in protein digestibility (*p* > 0.05). The energy digestibility, protein digestibility, and fat digestibility in the HE group were higher than those in the NE group (*p* < 0.05). The fat digestibility in the HF group was higher than that in the LF group (*p* < 0.05), but there was no difference in energy digestibility and protein digestibility (*p* > 0.05). There was an interaction between the temperature and energy level on energy digestibility (*p* < 0.05), and there was an interaction between the energy level and energy structure on protein digestibility (*p* < 0.05). At 10 °C, the energy digestibility in the HE group was higher than that in the NE group (*p* < 0.05). At 20 °C, the energy digestibility in the HE group was not different from that in the NE group (*p* > 0.05). In the NE group and he group, the energy digestibility at 10 °C was lower than that at 20 °C (*p* < 0.05). In the NE group, the protein digestibility of the HF group was higher than that of the LF group (*p* < 0.05). In the HE group, the protein digestibility of the HF group was not different from that of the LF group (*p* > 0.05). In the LF group, the protein digestibility of the HE group was higher than that of the NE group (*p* < 0.05). In the HF group, the protein digestibility of the HE group was not different from that of the NE group (*p* > 0.05) (Table 5).

### 3.4. Nitrogen Balance Test

When the temperature was 10 °C, urine nitrogen was higher than 20 °C (*p* < 0.05), the nitrogen deposition rate was lower than 20 °C (*p* < 0.05), and fecal nitrogen and nitrogen apparent digestibility had no difference (*p* > 0.05). Nitrogen apparent digestibility and the nitrogen deposition rate in the HE group were higher than those in the NE group (*p* < 0.05), fecal nitrogen was lower than that in the NE group (*p* < 0.05), and urinary nitrogen had no difference (*p* > 0.05). Urine nitrogen and fecal nitrogen in the HF group were higher than those in the LF group (*p* < 0.05), while nitrogen apparent digestibility and the nitrogen deposition rate in the HF group were lower than those in the LF group (*p* < 0.05) (Table 6).

### 3.5. O_2_ Consumption, CO_2_ Production, and RQ

At 10 °C, O_2_ consumption and CO_2_ production were higher than 20 °C (*p* < 0.05), and RQ was lower than 20 °C (*p* < 0.05). RQ in the HE group was lower than that in the NE group (*p* < 0.05), and there was no difference in O_2_ consumption and CO_2_ production (*p* > 0.05). RQ in the HF group was lower than that in the LF group (*p* < 0.05), and there was no difference in O_2_ consumption and CO_2_ production (*p* > 0.05). In terms of RQ, there was an interaction between temperature and energy structure (*p* < 0.05). At 10 °C, the RQ of the HF group was lower than that of the LF group (*p* < 0.05). At 20 °C, there was no difference between the HF group and LF group (*p* > 0.05). In the LF group and HF group, the RQ at 10 °C was lower than 20 °C (*p* < 0.05) (Table 7).

### 3.6. Blood Biochemistry Indicators

At 10 °C, the activities of ALB, TCHO, and Glu in blood were lower than 20 °C (*p* < 0.05), and the activity of bun was higher than 20 °C (*p* < 0.05). The Glu activity in the HE group was higher than that in the Le group (*p* < 0.05). The activities of TG, LDL, Glu, and HDL in the HF group were higher than those in the LF group (*p* < 0.05), while the activity of BUN in the HF group was lower than that in the LF group (*p* < 0.05). The temperature, feed energy level, and energy structure had no effect on blood TP activity (*p* > 0.05). In terms of bun and GLU activities, there was an interaction between temperature and energy structure (*p* < 0.05). At 10 °C and 20 °C, bun activity in the HF group was lower than that in the LF group (*p* < 0.05), and BUN activity at 10 °C was higher than that at 20 °C in LF and HF groups. There was no difference in GLU activity between the HF group and LF group at 10 °C (*p* > 0.05). At 20 °C, the GLU activity in the HF group was higher than that in the LF group (*p* < 0.05). In the LF group, there was no difference in GLU activity between 10 °C and 20 °C (*p* > 0.05). In the HF group, Glu activity at 10 °C was lower than that at 20 °C (*p* < 0.05) (Table 8 and Table 9).

### 3.7. Non-Targeted Metabolomics

A total of 1175 metabolites were identified after the combination of positive and negative ion modes in this experiment, and the number of metabolites identified by positive and negative ion modes were 653 and 522, respectively. After the difference analysis, an ANOVA *p* value < 0.01 was screened. After the combination of positive and negative ion modes, a total of 31 different metabolites were identified, as shown in Table 10. The top 20 items with the highest significance were selected according to the *p* value, and the bubble chart was drawn as shown in Figure 1. Among them, the relevant metabolism has 10 pathways at most, and the rest are related to human diseases, organic systems, environmental information processing, genetic information processing, and cellular processes. The most prevalent pathways are glycine, serine, and threonine metabolism; glycerophospholipid metabolism; biosynthesis of unsaturated fatty acids; etc.

## 4. Discussion

The temperature, feed energy level, and energy structure have different effects on pigs. The previous literature reported on the effects of each of the factors of the temperature, feed energy level, and energy structure individually on pigs [18,19,20,21]. However, there are some degrees of interaction effects among the three factors, which are rarely reported. The change in feed energy structure is particularly important for improving the energy utilization efficiency of pigs. Therefore, this study analyzed the effects of the temperature, feed energy level, and energy structure on the energy metabolism of Songliao black pigs; discussed the specific effects of three factors on them; and made recommendations for nutritional optimization protocols at different ambient temperatures at different stages of their growth.

This study found that the growth performance of the NT group was better than that of the LT group, which was consistent with the results of [19]. This is because in the cold environment, pigs’ oxygen consumption is increased and their shivering is more severe than that in the normal-temperature environment [22,23], resulting in increased energy loss and reduced energy deposition due to feeding the same energy feed. The growth performance of HE and HF groups was better than that of NE and LF groups, and the results of [5,24,25] were consistent with this study. It shows that high-energy and high-fat feed within a certain range is conducive to the growth of pigs. Pigs can deposit a higher-energy part into a body part, and in the fattening period, appropriately improve the level of feed fat, which is more conducive to the deposition of energy. This may be because the fat part in the feed can be directly deposited into body fat, thus eliminating the process of conversion from carbohydrates to fat and reducing energy consumption. Under the experimental conditions, the energy digestibility and fat digestibility of the LT group were lower than those of the NT group, indicating that when pigs were in a cold state, a large part of energy was used for heat production [26], resulting in the reduction in energy digestion and deposition efficiency. At the same time, fat was also decomposed for body heat production, resulting in the reduction in fat digestibility. The energy digestibility, protein digestibility, and fat digestibility of the HE group were higher than those of the NE group. When the energy intake of pigs was insufficient, the energy in the feed was preferentially used to maintain metabolism, and then used for digestion and deposition within the body, so the nutrient digestibility of the HE group was higher [27,28]. The fat digestibility of the HF group was higher than that of the LF group, but there was no difference in energy digestibility and protein digestibility, which was consistent with the research results of [29], indicating that the high-fat diet improved the energy deposition rate by improving the fat digestibility of pigs, but did not improve the energy digestibility and protein digestibility, which may be the direct deposition of fat in the feed, while in the LF group, the fat deposition of pigs may be formed by the conversion of carbohydrates in the feed, which will lead to energy loss and waste. Increasing the level of fat in feed, that is, changing the energy structure, can save energy. Under the low-temperature environment, the energy digestibility of the HE group was higher than that of the NE group (*p* < 0.05), but under the normal-temperature environment, the energy digestibility of the HE group and NE group had no difference (*p* < 0.05), indicating that under the low-temperature environment, the energy utilization efficiency of the high-energy diet was higher, and appropriately improving the energy level in the diet was conducive to the energy digestion of pigs. By analyzing the growth performance and nutrient digestibility of Songliao black pigs, we found that under the low-temperature environment, high-fat-level feed can improve the growth performance and fat digestibility of pigs, so as to improve the energy utilization efficiency. However, in terms of cost, too much addition of fats and oils may lead to insufficient cost savings and needs to be further examined and explored.

Energy in feed can be estimated as GE, DE, ME, and NE of pigs [30], including FE, UE, sedimentary energy, and total HI of pigs. In the cold state, the pig body trembles to resist the cold, and the energy required to maintain life increases [22,23]. More energy is wasted in feces and urine, resulting in lower energy utilization efficiency and less energy deposition. Fecal energy in the HE group was lower than that in the NE group, indicating that high-energy feed would lead to the increase in fecal energy of pigs and the decrease in estimated digestible energy. Fecal energy and deposition energy in the HF group were higher than those in the LF group, indicating that even though a high-oil diet would cause more fecal energy to be excluded from the body, compared with a low-oil diet, the energy efficiency that can be utilized and deposited by pigs is higher and the conversion is more direct [31,32]. The fecal energy of the HE group was lower than that of the NE group (*p* < 0.05), and the fecal energy of the HF group was higher than that of the LF group (*p* < 0.05), indicating that high-energy and high-fat diets can increase the proportion of digestible energy used by pigs in a low-temperature environment, so as to save dietary energy. In the low-temperature environment, OXFAT in the HE group was higher than that in the NE group (*p* < 0.05), and OXFAT in the HF group was higher than that in the LF group (*p* < 0.05), indicating that pigs fed a high-energy diet used more fat to release heat in the low-temperature environment, which may be because the daily energy intake of pigs was relatively sufficient at this time, and the fat or body fat in the feed could be used to produce heat. In a low-temperature environment, pigs fed a high-fat diet can directly use the fat in the diet to produce heat, thus reducing the loss and consumption of energy. Under the low-temperature environment, the O_2_ consumption and CO_2_ output of Songliao black pigs increased, which was consistent with the results of [22,23]. The respiratory quotient was lower than the normal-temperature state, indicating that more fat was used for energy supply at this time. At this time, appropriately increasing the proportion of fat in the feed was conducive to direct conversion into pig fat, so as to save energy. The RQ of the HE group and HF group were lower than that of the LE and LF group (*p* < 0.05), indicating that the proportion of fat energy supply increased at this time. Increasing the proportion of fat in the feed leads to an increase in the proportion of energy supplied by fat, which indicates that in the fattening pig feed, increasing the proportion of fat can directly transform into the deposition of body fat, which has the effect of saving energy. Under the low-temperature environment, RQ of the HF group was lower than that of the LF group (*p* < 0.05), which indicated that feeding pigs with the high-fat-level diet under the low-temperature environment was conducive to direct energy supply of fat and energy saving. This evidence proves that the fat in the feed can be directly transformed into body fat deposition, and in the low-temperature environment, the higher the oil level in the feed, the higher the proportion of fat energy supply, and the efficiency of energy supply will be far greater than that of carbohydrate and protein energy supply, thus saving energy sources and raw materials. At the same time, increasing the fat level in the diet will reduce the CO_2_ released from the pork chops, thus playing a role in protecting the environment. However, whether the addition of fat to feed leads to an increase in nitrogen emissions remains to be demonstrated in more trials.

Blood urea nitrogen reflects the digestibility of protein in animals [33]. In a low-temperature setting, the bun activity in Songliao black pigs markedly surpasses that observed at standard temperatures, indicating enhanced protein digestibility compared to normal environmental conditions. LDL is the main source of cholesterol accumulation [34]. Increasing dietary fat levels leads to an increase in LDL, which may mean an increase in cholesterol in pigs. ALB is crucial for sustaining osmotic pressure and facilitating the transport of various substances within the bloodstream [35]. Albumin activity at 10 °C is lower than 20 °C, indicating that at 10 °C, pigs are in a “sub-health” state relative to 20 °C. Glucose, triglyceride, total protein, albumin, and urea concentrations are frequently utilized as primary biochemical markers in blood analyses. It is the most commonly used blood biochemical index to evaluate the metabolism of carbohydrates, fat, and protein. Elevated levels within the normal range of these concentrations indicate improved energy and protein nutritional status [36]. In the study, dietary fat augmentation enhanced the activities of TG, LDL, and GLU, suggesting that a higher fat intake can beneficially affect the energy and protein nutritional status of pigs. This enhancement in GLU levels is likely tied to the accelerated fattening rate and increased backfat thickness in pigs. The synthesis of TG predominantly relies on glycerol and fatty acids derived from GLU [37]. Therefore, increasing the dietary fat level may improve the fattening speed and backfat thickness of pigs. In the normal-temperature setting, the enzymatic activity of GLU in the HF group surpassed that in the LF group, marked by a statistical difference (*p* < 0.05). This suggests that elevating dietary fat levels can enhance carbohydrate metabolism in pigs, potentially speeding up fattening and increasing backfat thickness. Conversely, under low-temperature conditions, differences in GLU activity between the HF and LF groups were not statistically different (*p* > 0.05), indicating that temperature may influence the metabolic impact of dietary fat. Through the analysis of blood biochemical indexes of Songliao black pigs, it was proved that improving the oil level of the diet could improve the energy utilization efficiency, and was beneficial to the fattening speed and backfat growth of Songliao black pigs. However, the addition of fat to feed may have an impact on animal health conditions that needs to be further examined.

Under the experimental conditions, 31 kinds of differential metabolites were screened according to a *p* value < 0.01. Most of them were related to the pathway of metabolism, mainly affecting the pathway of glycine, serine, and threonine metabolism; glycosphospholipid metabolism; and biosynthesis of unsaturated fatty acids, which showed that the change in dietary energy composition affected the lipid metabolism and amino acid metabolism and synthesis of Songliao black pigs, which showed the change in growth performance and nutrient digestion and metabolism. It was feasible to affect the growth of Songliao black pigs by changing the energy composition of the diet. At the same time, it also affected the pathways related to human diseases, including choline metabolism in cancer, African trypanosomias, and chemical carcinogenesis—receptor activation and central carbon metabolism in cancer. The change in ambient temperature may affect the pathways related to environmental information processing, including cholinergic synapses and serotronergic synapses. Finally, the processing of this experiment also affected the related pathways of genetic information processing, which may have a certain impact on the genetic traits of Songliao black pigs. The results showed that the environmental temperature, feed energy level, and energy structure had an impact on the metabolism of Songliao black pigs, which had an impact on amino acid metabolism and fatty acid metabolism, and affected their biological system and genetic traits. The change in environmental temperature may also affect the processing of environmental information of Songliao black pigs.

In this experiment, the sample size was small due to the expensive use of precision instruments, but the data were well parallelized. The effects of the ambient temperature, feed energy level, and energy structure on growth performance, nutrient digestibility, and energy metabolism of Songliao black fattening pigs were investigated. Since Songliao black pigs are lean pigs, the theoretical dietary recommendations can be improved in lean pig breeds, and more experimental studies are needed to collect evidence for the further precision of feed nutritional formulations.

## 5. Conclusions

The energy metabolism of Songliao black finishing pigs is not only affected by the environmental temperature and feed energy level, but also by the feed energy structure, and there are interactions among these three factors. Appropriately increasing the energy level of the diet and improving the energy structure of the feed (increasing the oil level) will be conducive to the growth of Songliao black finishing pigs and improve their energy utilization efficiency, while reducing the emissions of CO_2_ and other pollutants, and these changes are more obvious in a cold environment. In summary, our results demonstrate the need to provide more suitable energy nutrition supply solutions for the changes in the energy metabolic pathways of the animal organism under different ambient temperature conditions, and provide a theoretical basis for a more accurate match between energy supply and demand, and the establishment of a swine energy nutrition system that makes efficient use of feed resources and reduces the emission of pollutants.

## Figures and Tables

**Figure 1 animals-14-03061-f001:**
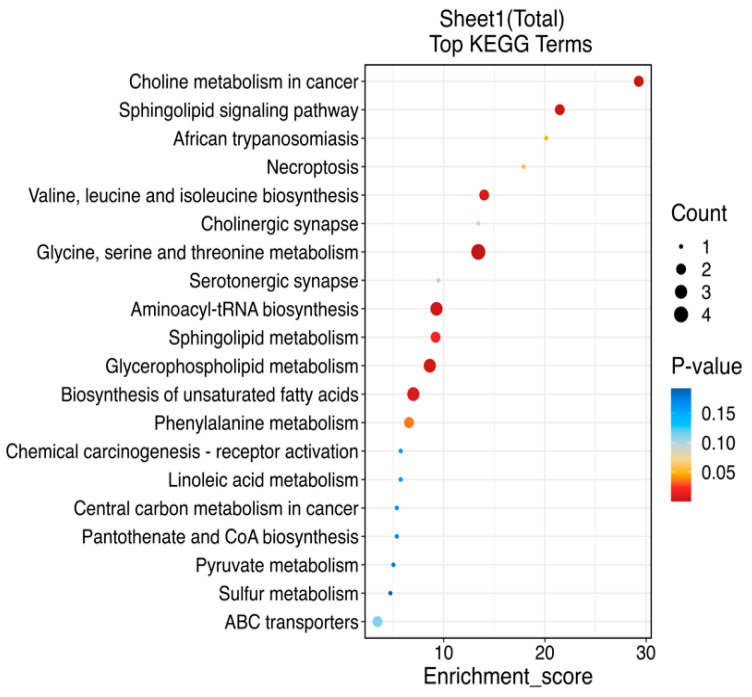
KEGG enrichment pathway map through metabolomics analysis in plasma samples from Songliao black fattening pigs at different ambient temperatures, energy levels, and energy sources.

**Table 1 animals-14-03061-t001:** Composition and nutrient levels of diet (60–100 kg).

Item	NELF	NEHF	HELF	HEHF
Corn	57.46%	67.00%	45.08%	67.90%
Corn starch	16.34%	0.50%	32.47%	3.48%
Wheat bran	8.68%	2.76%	0.10%	6.61%
Soybean meal	13.15%	13.31%	17.90%	11.95%
Monocalcium phosphate	0.48%	0.56%	0.61%	0.50%
Stone flour	0.99%	0.93%	0.87%	0.98%
Salt	0.22%	0.22%	0.23%	0.21%
Soybean oil	0.78%	5.74%	1.66%	6.62%
Lysine	0.32%	0.32%	0.25%	0.34%
Methionine	0.06%	0.05%	0.08%	0.05%
Threonine	0.11%	0.10%	0.09%	0.11%
Tryptophan	0.02%	0.02%	0.01%	0.02%
Valine	0.05%	0.04%	0.09%	0.05%
Alpha-cellulose	0.84%	7.95%	0.06%	0.68%
Premix	0.50%	0.50%	0.50%	0.50%
Total	100.00%	100.00%	100.00%	100.00%
Nutritional level				
Digestible energy (MJ/kg)	14.02	14.02	15.14	15.14
Digestible crude protein (%)	10.06	10.06	10.06	10.06
Lysine (%)	0.70	0.70	0.70	0.70
Tryptophan (%)	0.12	0.12	0.12	0.12
Methionine + cystine (%)	0.40	0.40	0.40	0.40
Threonine (%)	0.45	0.45	0.45	0.45
Calcium (%)	0.56	0.56	0.56	0.56
Available phosphorus (%)	0.19	0.19	0.19	0.19
Nitrogen-free extractives (%)	66.03	53.97	69.48	59.37
Crude fat (%)	3.46	8.40	3.65	9.43

The premix provided the following per kg of diets: VA, 1350 IU; VD3, 160 IU; VE, 14 IU; VK, 0.5 mg; choline, 0.40 g; vitamin B6, 1 mg; vitamin B12, 6.0 μg; biotin, 0.07 mg; folic acid, 0.3 mg; nicotinic acid, 10 mg; pantothenic acid, 8 mg; thiamine, 1.5 mg; riboflavin, 2 mg; Cu, 3.5 mg; I, 0.14 mg; Fe, 50 mg; Mn, 2 mg; Se, 0.25 mg; Zn, 50 mg. Values of digestible energy were calculated from data provided by the Feed Database in China (2019). NELF = normal energy, low fat; NEHF = normal energy, high fat; HELF = high energy, low fat; HEHF = high energy, high fat.

**Table 2 animals-14-03061-t002:** Effects on growth performance and nutrient digestibility of Songliao black fattening pigs.

Temperature	Energy Level	Energy Structure	Average Daily Feed Intake (kg/d)	Average Daily Weight Gain (kg/d)	Feed-to-Gain Ratio
LT	NE	LF	2.53 ± 0.04 ^ab^	0.59 ± 0.05 ^e^	4.34 ± 0.41 ^a^
		HF	2.63 ± 0.15 ^a^	0.73 ± 0.13 ^de^	3.67 ± 0.51 ^b^
	HE	LF	2.44 ± 0.21 ^b^	0.69 ± 0.07 ^de^	3.61 ± 0.60 ^bc^
		HF	2.64 ± 0.02 ^a^	0.77 ± 0.10 ^cd^	3.46 ± 0.45 ^bc^
NT	NE	LF	2.45 ± 0.03 ^b^	0.82 ± 0.09 ^cd^	3.01 ± 0.32 ^cd^
		HF	2.51 ± 0.33 ^ab^	1.00 ± 0.11 ^ab^	2.53 ± 0.29 ^de^
	HE	LF	2.43 ± 0.26 ^b^	0.91 ± 0.13 ^bc^	2.71 ± 0.39 ^de^
		HF	2.52 ± 0.44 ^ab^	1.09 ± 0.04 ^a^	2.32 ± 0.11 ^e^
Influencing factors					
Temperature	LT		2.56 ± 0.14 ^a^	0.69 ± 0.11 ^b^	3.77 ± 0.56 ^a^
	NT		2.48 ± 0.05 ^b^	0.96 ± 0.13 ^a^	2.64 ± 0.37 ^b^
Energy level	NE		2.53 ± 0.10 ^a^	0.79 ± 0.18 ^b^	3.39 ± 0.79 ^a^
	HE		2.51 ± 0.13 ^a^	0.88 ± 0.17 ^a^	3.02 ± 0.67 ^b^
Energy structure	LF		2.46 ± 0.11 ^b^	0.75 ± 0.15 ^b^	3.42 ± 0.76 ^a^
	HF		2.57 ± 0.09 ^a^	0.89 ± 0.18 ^a^	2.99 ± 0.69 ^b^
*p* value					
Temperature			0.025	<0.001	<0.001
Energy level			0.495	0.026	0.020
Energy structure			0.004	<0.001	<0.001
Temperature × Energy level			0.591	0.795	0.469
Temperature × Energy structure			0.320	0.357	0.929
Energy level × Energy structure			0.369	0.630	0.301
Temperature × Energy level × Energy structure			0.556	0.738	0.474

Data in the same column with different superscript letters indicate differences (*p* < 0.05), while data with the same superscript letter indicate no differences (*p* > 0.05). F:G = feed-to-gain ratio. LT = low temperature; NT = normal temperature; NE = normal energy; HE = high energy; LF = low fat; HF = high fat.

**Table 3 animals-14-03061-t003:** Effects on energy partition of Songliao black fattening pigs.

Temperature	Energy Level	Energy Structure	Total Energy (MJ/d)	Total Heat Production (MJ/d)	Manure Energy (MJ/d)	Urine Energy (MJ/d)	Sedimentation Energy (MJ/d)
LT	NE	LF	39.55 ± 0.71 ^de^	22.05 ± 0.63 ^a^	9.17 ± 1.53 ^b^	1.03 ± 0.09 ^a^	7.30 ± 2.29 ^c^
		HF	46.89 ± 4.31 ^a^	21.17 ± 0.72 ^abc^	11.55 ± 1.25 ^a^	1.02 ± 0.27 ^a^	13.16 ± 2.88 ^b^
	HE	LF	41.50 ± 2.26 ^cd^	22.13 ± 1.18 ^a^	6.83 ± 1.46 ^c^	1.16 ± 0.08 ^a^	11.38 ± 3.61 ^b^
		HF	44.95 ± 0.41 ^ab^	21.73 ± 1.30 ^ab^	8.54 ± 1.72 ^b^	1.01 ± 0.18 ^a^	13.68 ± 2.56 ^b^
NT	NE	LF	38.21 ± 0.49 ^e^	19.86 ± 1.12 ^bc^	3.88 ± 0.81 ^d^	0.85 ± 0.29 ^a^	13.62 ± 1.13 ^b^
		HF	42.40 ± 0.56 ^bc^	19.38 ± 1.95 ^c^	4.24 ± 1.06 ^d^	0.84 ± 0.11 ^a^	17.94 ± 2.38 ^a^
	HE	LF	38.68 ± 0.41 ^e^	19.86 ± 1.01 ^bc^	3.48 ± 0.18 ^d^	0.96 ± 0.26 ^a^	14.37 ± 1.32 ^b^
		HF	42.95 ± 0.76 ^bc^	19.36 ± 1.67 ^c^	4.05 ± 0.33 ^d^	0.83 ± 0.22 ^a^	18.70 ± 1.28 ^a^
Influencing factors							
Temperature	LT		43.22 ± 3.70 ^a^	21.77 ± 0.98 ^a^	9.02 ± 2.20 ^a^	1.05 ± 0.17 ^a^	11.38 ± 3.65 ^b^
	NT		40.56 ± 2.26 ^b^	19.62 ± 1.36 ^b^	3.91 ± 0.69 ^b^	0.87 ± 0.21 ^b^	16.16 ± 2.68 ^a^
Energy level	NE		41.76 ± 3.97 ^a^	20.61 ± 1.54 ^a^	7.21 ± 3.54 ^a^	0.93 ± 0.21 ^a^	13.01 ± 4.40 ^a^
	HE		42.02 ± 2.60 ^a^	20.77 ± 1.69 ^a^	5.73 ± 2.36 ^b^	0.99 ± 0.21 ^a^	14.53 ± 3.47 ^a^
Energy structure	LF		39.48 ± 1.70 ^b^	20.98 ± 1.46 ^a^	5.84 ± 2.60 ^b^	1.00 ± 0.21 ^a^	11.67 ± 3.51 ^b^
	HF		44.30 ± 2.70 ^a^	20.41 ± 1.72 ^a^	7.09 ± 3.41 ^a^	0.92 ± 0.20 ^a^	15.87 ± 3.98 ^a^
*p* value							
Temperature			<0.001	<0.001	<0.001	0.016	<0.001
Energy level			0.687	0.727	<0.001	0.432	0.077
Energy structure			<0.001	0.219	<0.001	0.316	<0.001
Temperature × Energy level			0.696	0.717	<0.001	0.952	0.358
Temperature × Energy structure			0.364	0.866	0.068	0.955	0.884
Energy level × Energy structure			0.146	0.803	0.782	0.380	0.293
Temperature × Energy level × Energy structure			0.130	0.787	0.602	0.931	0.291

Data in the same column with different superscript letters indicate differences (*p* < 0.05), while data with the same superscript letter indicate no differences (*p* > 0.05). LT = low temperature; NT = normal temperature; NE = normal energy; HE = high energy; LF = low fat; HF = high fat.

**Table 4 animals-14-03061-t004:** Effects on metabolism indicators and oxidation of energy-providing nutrients of Songliao black fattening pigs.

Temperature	Energy Level	Energy Structure	Deposition of Protein (MJ/d)	Deposition of Fat (MJ/d)	OXCHO (MJ/d)	OXPRO (MJ/d)	OXFAT (MJ/d)
LT	NE	LF	4.80 ± 0.26 ^b^	2.50 ± 2.12 ^c^	18.49 ± 0.63 ^a^	1.89 ± 0.08 ^a^	1.70 ± 0.06 ^c^
		HF	4.81 ± 0.58 ^b^	8.34 ± 2.41 ^b^	16.09 ± 1.01 ^b^	2.03 ± 0.27 ^a^	3.08 ± 0.68 ^b^
	HE	LF	4.99 ± 0.64 ^ab^	4.48 ± 1.88 ^c^	17.64 ± 1.15 ^ab^	1.75 ± 0.17 ^a^	2.77 ± 0.11 ^b^
		HF	5.02 ± 0.32 ^ab^	8.54 ± 1.97 ^b^	15.93 ± 1.70 ^b^	1.98 ± 0.29 ^a^	3.85 ± 0.60 ^a^
NT	NE	LF	5.36 ± 0.16 ^ab^	8.26 ± 1.01 ^b^	17.47 ± 1.08 ^ab^	1.30 ± 0.13 ^b^	1.10 ± 0.09 ^c^
		HF	5.34 ± 0.15 ^ab^	12.60 ± 2.50 ^a^	16.72 ± 2.38 ^ab^	1.39 ± 0.14 ^b^	1.29 ± 0.43 ^c^
	HE	LF	5.58 ± 0.23 ^a^	8.79 ± 1.31 ^b^	17.49 ± 1.05 ^ab^	1.25 ± 0.13 ^b^	1.14 ± 0.09 ^c^
		HF	5.49 ± 0.23 ^a^	13.21 ± 1.25 ^a^	16.47 ± 1.45 ^ab^	1.34 ± 0.14 ^b^	1.57 ± 0.54 ^c^
Influencing factors							
Temperature	LT		4.91 ± 0.44 ^b^	5.89 ± 3.31 ^b^	17.03 ± 1.54 ^a^	1.91 ± 0.23 ^a^	2.85 ± 0.90 ^a^
	NT		5.44 ± 0.20 ^a^	10.72 ± 2.70 ^a^	17.04 ± 1.49 ^a^	1.32 ± 0.13 ^b^	1.27 ± 0.36 ^b^
Energy level	NE		5.08 ± 0.41 ^a^	7.93 ± 4.15 ^a^	17.19 ± 15.83 ^a^	1.65 ± 0.35 ^a^	1.79 ± 0.88 ^b^
	HE		5.27 ± 0.45 ^a^	9.08 ± 3.48 ^a^	16.88 ± 1.42 ^a^	1.58 ± 0.35 ^a^	2.33 ± 1.16 ^a^
Energy structure	LF		5.18 ± 0.46 ^a^	6.11 ± 3.12 ^b^	17.77 ± 0.99 ^a^	1.55 ± 0.31 ^b^	1.68 ± 0.70 ^b^
	HF		5.16 ± 0.42 ^a^	10.82 ± 2.98 ^a^	16.30 ± 1.56 ^b^	1.68 ± 0.39 ^a^	2.45 ± 1.21 ^a^
*p* value							
Temperature			<0.001	<0.001	0.992	<0.001	<0.001
Energy level			0.152	0.244	0.538	0.260	<0.001
Energy structure			0.881	<0.001	<0.001	0.046	<0.001
Temperature × Energy level			0.950	0.713	0.693	0.761	0.014
Temperature × Energy structure			0.771	0.681	0.250	0.480	<0.001
Energy level × Energy structure			0.899	0.547	0.832	0.730	0.917
Temperature × Energy level × Energy structure			0.860	0.506	0.634	0.747	0.350

Data in the same column with different superscript letters indicate differences (*p* < 0.05), while data with the same superscript letter indicate no differences (*p* > 0.05). OXCHO = carbohydrate oxidation; OXPRO = protein oxidation; OXFAT = fat oxidation. LT = low temperature; NT = normal temperature; NE = normal energy; HE = high energy; LF = low fat; HF = high fat.

**Table 5 animals-14-03061-t005:** Effects on nutrient digestibility of Songliao black fattening pigs.

Temperature	Energy Level	Energy Structure	Energy Digestibility (%)	Protein Digestibility (%)	Fat Digestibility (%)
LT	NE	LF	76.80 ± 3.90 ^c^	85.09 ± 4.24 ^b^	61.55 ± 1.97 ^d^
		HF	75.40 ± 0.71 ^c^	89.58 ± 4.85 ^a^	69.30 ± 4.05 ^c^
	HE	LF	83.51 ± 3.65 ^b^	90.36 ± 2.01 ^a^	66.21 ± 2.81 ^c^
		HF	78.80 ± 3.70 ^c^	89.98 ± 1.86 ^a^	75.01 ± 3.16 ^b^
NT	NE	LF	89.85 ± 2.16 ^a^	85.00 ± 1.85 ^b^	77.21 ± 3.90 ^b^
		HF	90.02 ± 2.39 ^a^	86.32 ± 1.65 ^ab^	84.75 ± 3.60 ^a^
	HE	LF	90.99 ± 0.49 ^a^	89.74 ± 1.65 ^a^	79.45 ± 2.06 ^b^
		HF	90.57 ± 0.62 ^a^	86.96 ± 1.77 ^ab^	88.83 ± 2.86 ^a^
Influencing factors					
Temperature	LT		78.63 ± 4.31 ^b^	88.75 ± 3.83 ^a^	68.02 ± 5.76 ^b^
	NT		90.36 ± 1.55 ^a^	87.00 ± 2.37 ^a^	82.56 ± 5.49 ^a^
Energy level	NE		83.02 ± 7.52 ^b^	86.50 ± 3.63 ^b^	73.20 ± 9.48 ^b^
	HE		85.97 ± 5.77 ^a^	89.26 ± 2.15 ^a^	77.37 ± 8.77 ^a^
Energy structure	LF		85.29 ± 6.40 ^a^	87.55 ± 3.52 ^a^	71.10 ± 8.09 ^b^
	HF		83.70 ± 7.22 ^a^	88.21 ± 3.05 ^a^	79.47 ± 8.55 ^a^
*p* value					
Temperature			<0.001	0.086	<0.001
Energy level			<0.001	<0.001	<0.001
Energy structure			0.095	0.506	<0.001
Temperature × Energy level			0.031	0.942	0.370
Temperature × Energy structure			0.123	0.168	0.934
Energy level × Energy structure			0.298	0.031	0.520
Temperature × Energy level × Energy structure			0.465	0.845	0.864

Data in the same column with different superscript letters indicate differences (*p* < 0.05), while data with the same superscript letter indicate no differences (*p* > 0.05). LT = low temperature; NT = normal temperature; NE = normal energy; HE = high energy; LF = low fat; HF = high fat.

**Table 6 animals-14-03061-t006:** Effects on N balance of Songliao black fattening pigs.

Temperature	Energy Level	Energy Structure	Ingested Nitrogen (g/d)	Urine Nitrogen (g/d)	Fecal Nitrogen (g/d)	Nitrogen Apparent Digestibility (%)	Nitrogen Deposition Rate (%)
LT	NE	LF	55.99 ± 1.00 ^ab^	16.39 ± 0.73 ^a^	7.42 ± 1.57 ^a^	86.74 ± 2.88 ^bc^	57.46 ± 2.26 ^cd^
		HF	58.05 ± 3.37 ^a^	17.61 ± 2.37 ^a^	8.18 ± 1.33 ^a^	85.83 ± 2.73 ^c^	55.47 ± 4.21 ^d^
	HE	LF	53.92 ± 4.62 ^b^	15.18 ± 1.48 ^a^	5.26 ± 1.45 ^b^	90.37 ± 2.01 ^a^	61.96 ± 4.06 ^bc^
		HF	58.26 ± 0.53 ^a^	17.16 ± 2.56 ^a^	7.45 ± 0.91 ^a^	87.21 ± 1.57 ^bc^	57.76 ± 3.73 ^cd^
NT	NE	LF	54.09 ± 0.69 ^b^	11.33 ± 1.15 ^b^	6.82 ± 0.22 ^ab^	87.39 ± 0.56 ^abc^	66.45 ± 1.95 ^ab^
		HF	55.53 ± 0.74 ^ab^	12.11 ± 1.23 ^b^	7.60 ± 0.94 ^a^	86.32 ± 1.66 ^c^	64.51 ± 1.20 ^b^
	HE	LF	53.74 ± 0.57 ^b^	10.84 ± 1.11 ^b^	5.46 ± 0.84 ^b^	89.82 ± 1.65 ^ab^	69.65 ± 2.49 ^a^
		HF	55.66 ± 2.52 ^ab^	11.64 ± 1.24 ^b^	7.23 ± 1.03 ^a^	87.02 ± 1.70 ^bc^	66.08 ± 2.09 ^ab^
Influencing factors							
Temperature	LT		56.56 ± 3.18 ^a^	16.58 ± 1.97 ^a^	7.08 ± 1.65 ^a^	87.54 ± 2.75 ^a^	58.16 ± 4.08 ^b^
	NT		54.76 ± 1.11 ^b^	11.48 ± 1.16 ^b^	6.78 ± 1.11 ^a^	87.64 ± 1.90 ^a^	66.67 ± 2.63 ^a^
Energy level	NE		55.92 ± 2.20 ^a^	14.36 ± 3.08 ^a^	7.50 ± 1.13 ^a^	86.57 ± 2.03 ^b^	60.97 ± 5.32 ^b^
	HE		55.39 ± 2.85 ^a^	13.71 ± 3.07 ^a^	6.35 ± 1.41 ^b^	88.60 ± 2.20 ^a^	63.86 ± 5.42 ^a^
Energy structure	LF		54.44 ± 2.35 ^b^	13.43 ± 2.68 ^b^	6.24 ± 1.39 ^b^	88.58 ± 2.37 ^a^	63.88 ± 5.38 ^a^
	HF		56.88 ± 2.10 ^a^	14.63 ± 3.35 ^a^	7.62 ± 1.02 ^a^	86.59 ± 1.85 ^b^	60.95 ± 5.35 ^b^
*p* value							
Temperature			0.025	<0.001	0.456	0.885	<0.001
Energy level			0.495	0.260	<0.001	<0.001	0.010
Energy structure			<0.001	0.046	<0.001	<0.01	0.010
Temperature × Energy level			0.591	0.094	0.464	0.507	0.634
Temperature × Energy structure			0.320	0.480	0.796	0.947	0.871
Energy level × Energy structure			0.369	0.730	0.137	0.165	0.365
Temperature × Energy level × Energy structure			0.556	0.747	0.780	0.853	0.890

Data in the same column with different superscript letters indicate differences (*p* < 0.05), while data with the same superscript letter indicate no differences (*p* > 0.05). LT = low temperature; NT = normal temperature; NE = normal energy; HE = high energy; LF = low fat; HF = high fat.

**Table 7 animals-14-03061-t007:** Effects on RQ, O_2_ consumption, and CO_2_ emission of Songliao black fattening pigs.

Temperature	Energy Level	Energy Structure	O_2_ Consumption (L/d)	CO_2_ Emission (L/d)	RQ
LT	NE	LF	1054.80 ± 29.69 ^a^	1011.60 ± 29.69 ^a^	0.96 ± 0.00 ^a^
		HF	1018.80 ± 34.03 ^ab^	954.00 ± 37.87 ^ab^	0.94 ± 0.01 ^bc^
	HE	LF	1062.00 ± 55.62 ^a^	1004.40 ± 55.62 ^a^	0.95 ± 0.01 ^b^
		HF	1047.60 ± 60.38 ^a^	972.00 ± 67.03 ^ab^	0.93 ± 0.01 ^c^
NT	NE	LF	946.80 ± 53.07 ^b^	918.00 ± 53.07 ^ab^	0.97 ± 0.00 ^a^
		HF	925.20 ± 90.60 ^b^	892.80 ± 96.24 ^b^	0.97 ± 0.01 ^a^
	HE	LF	946.80 ± 47.58 ^b^	918.00 ± 47.58 ^ab^	0.97 ± 0.00 ^a^
		HF	925.20 ± 80.07 ^b^	889.20 ± 75.63 ^b^	0.96 ± 0.02 ^a^
Influencing factors					
Temperature	LT		1045.80 ± 45.19 ^a^	985.50 ± 50.70 ^a^	0.94 ± 0.01 ^b^
	NT		936.00 ± 63.75 ^a^	904.50 ± 64.87 ^b^	0.97 ± 0.01 ^a^
Energy level	NE		986.40 ± 74.55 ^a^	944.10 ± 70.73 ^a^	0.96 ± 0.01 ^a^
	HE		995.40 ± 83.28 ^a^	945.90 ± 72.62 ^a^	0.95 ± 0.02 ^b^
Energy structure	LF		1002.60 ± 71.69 ^a^	963.00 ± 63.07 ^a^	0.96 ± 0.01 ^a^
	HF		979.20 ± 84.29 ^a^	927.00 ± 74.89 ^a^	0.95 ± 0.02 ^b^
*p* value					
Temperature			<0.001	<0.001	<0.001
Energy level			0.673	0.934	0.019
Energy structure			0.278	0.109	<0.001
Temperature × Energy level			0.673	0.869	0.050
Temperature × Energy structure			0.933	0.681	0.019
Energy level × Energy structure			0.800	0.805	0.820
Temperature × Energy level × Energy structure			0.800	0.742	0.498

Data in the same column with different superscript letters indicate differences (*p* < 0.05), while data with the same superscript letter indicate no differences (*p* > 0.05). RQ = respiratory quotient. LT = low temperature; NT = normal temperature; NE = normal energy; HE = high energy; LF = low fat; HF = high fat.

**Table 8 animals-14-03061-t008:** Effects on biochemical marker assays in serum of Songliao black fattening pigs.

Temperature	Energy Level	Energy Structure	TP (g/L)	ALB (g/L)	TG (mmol/L)	GLU (mmol/L)
LT	NE	LF	65.05 ± 0.92 ^a^	23.80 ± 1.07 ^c^	0.27 ± 0.03 ^c^	4.28 ± 0.13 ^c^
		HF	64.99 ± 1.21 ^a^	24.67 ± 0.98 ^bc^	0.32 ± 0.04 ^abc^	4.38 ± 0.19 ^bc^
	HE	LF	65.66 ± 0.89 ^a^	25.70 ± 0.50 ^b^	0.30 ± 0.03 ^bc^	4.34 ± 0.15 ^c^
		HF	65.70 ± 0.58 ^a^	24.21 ± 0.85 ^c^	0.32 ± 0.03 ^ab^	4.51 ± 0.12 ^bc^
NT	NE	LF	65.61 ± 0.78 ^a^	27.00 ± 0.71 ^a^	0.30 ± 0.03 ^bc^	4.37 ± 0.15 ^bc^
		HF	65.52 ± 0.74 ^a^	26.96 ± 0.77 ^a^	0.32 ± 0.03 ^ab^	4.61 ± 0.20 ^b^
	HE	LF	65.90 ± 0.97 ^a^	27.31 ± 0.32 ^a^	0.31 ± 0.03 ^abc^	4.46 ± 0.16 ^bc^
		HF	65.45 ± 0.52 ^a^	26.85 ± 0.25 ^a^	0.35 ± 0.01 ^a^	5.05 ± 0.19 ^a^
Influencing factors						
Temperature	LT		65.35 ± 0.90 ^a^	24.59 ± 1.07 ^b^	0.30 ± 0.03 ^a^	4.38 ± 0.16 ^b^
	NT		65.62 ± 0.71 ^a^	27.03 ± 0.53 ^a^	0.32 ± 0.03 ^a^	4.62 ± 0.31 ^a^
Energy level	NE		65.29 ± 0.88 ^a^	25.61 ± 1.66 ^a^	0.30 ± 0.04 ^a^	4.41 ± 0.20 ^b^
	HE		65.68 ± 0.70 ^a^	26.02 ± 1.33 ^a^	0.32 ± 0.03 ^a^	4.59 ± 0.31 ^a^
Energy structure	LF		65.55 ± 0.86 ^a^	25.95 ± 1.56 ^a^	0.29 ± 0.03 ^b^	4.36 ± 0.15 ^b^
	HF		65.41 ± 0.77 ^a^	25.67 ± 1.46 ^a^	0.33 ± 0.03 ^a^	4.64 ± 0.30 ^a^
*p* value						
Temperature			0.382	<0.001	0.105	<0.001
Energy level			0.213	0.128	0.083	0.005
Energy structure			0.648	0.300	0.003	<0.001
Temperature × Energy level			0.376	0.250	0.758	0.149
Temperature × Energy structure			0.677	0.911	0.853	0.025
Energy level × Energy structure			0.833	0.013	0.758	0.082
Temperature × Energy level × Energy structure			0.708	0.074	0.499	0.240

Data in the same column with different superscript letters indicate differences (*p* < 0.05), while data with the same superscript letter indicate no differences (*p* > 0.05). TP = total protein; ALB = albumin; TG = triglyceride; GLU = glucose. LT = low temperature; NT = normal temperature; NE = normal energy; HE = high energy; LF = low fat; HF = high fat.

**Table 9 animals-14-03061-t009:** Effects on biochemical marker assays in serum of Songliao black fattening pigs.

Temperature	Energy Level	Energy Structure	LDL (mmol/L)	HDL (mmol/L)	TCHO (mmol/L)	BUN (mmol/L)
LT	NE	LF	0.91 ± 0.06 ^b^	0.89 ± 0.06 ^b^	2.07 ± 0.09 ^b^	3.14 ± 0.06 ^ab^
		HF	0.98 ± 0.07 ^ab^	0.98 ± 0.08 ^ab^	2.07 ± 0.08 ^b^	3.04 ± 0.09 ^b^
	HE	LF	0.93 ± 0.07 ^ab^	0.89 ± 0.10 ^b^	2.08 ± 0.09 ^b^	3.22 ± 0.09 ^a^
		HF	1.01 ± 0.06 ^ab^	1.01 ± 0.08 ^a^	2.19 ± 0.08 ^b^	3.04 ± 0.08 ^b^
NT	NE	LF	0.96 ± 0.07 ^ab^	0.94 ± 0.06 ^ab^	2.33 ± 0.14 ^a^	3.02 ± 0.12 ^b^
		HF	0.98 ± 0.07 ^ab^	0.97 ± 0.03 ^ab^	2.40 ± 0.08 ^a^	2.64 ± 0.10 ^c^
	HE	LF	0.97 ± 0.07 ^ab^	0.97 ± 0.06 ^ab^	2.33 ± 0.10 ^a^	3.07 ± 0.09 ^b^
		HF	1.02 ± 0.06 ^a^	1.05 ± 0.06 ^a^	2.40 ± 0.08 ^a^	2.75 ± 0.10 ^c^
Influencing factors						
Temperature	LT		0.96 ± 0.07 ^a^	0.94 ± 0.09 ^a^	2.10 ± 0.09 ^b^	3.11 ± 0.11 ^a^
	NT		0.98 ± 0.06 ^a^	0.98 ± 0.06 ^a^	2.37 ± 0.10 ^a^	2.87 ± 0.21 ^b^
Energy level	NE		0.95 ± 0.07 ^a^	0.94 ± 0.06 ^a^	2.22 ± 0.18 ^a^	2.96 ± 0.21 ^a^
	HE		0.98 ± 0.07 ^a^	0.98 ± 0.09 ^a^	2.25 ± 0.15 ^a^	3.02 ± 0.19 ^a^
Energy structure	LF		0.94 ± 0.06 ^b^	0.92 ± 0.07 ^b^	2.21 ± 0.16 ^a^	3.11 ± 0.11 ^a^
	HF		1.00 ± 0.06 ^a^	1.00 ± 0.07 ^a^	2.26 ± 0.16 ^a^	2.87 ± 0.20 ^b^
*p* value						
Temperature			0.301	0.082	<0.001	<0.001
Energy level			0.254	0.146	0.337	0.073
Energy structure			0.017	0.003	0.089	<0.001
Temperature × Energy level			0.912	0.507	0.355	0.597
Temperature × Energy structure			0.412	0.335	0.765	0.004
Energy level × Energy structure			0.621	0.445	0.374	0.850
Temperature × Energy level × Energy structure			0.912	0.798	0.394	0.294

Data in the same column with different superscript letters indicate differences (*p* < 0.05), while data with the same superscript letter indicate no differences (*p* > 0.05). LDL = low-density lipoprotein; HDL = high-density lipoprotein; TCHO = total cholesterol; BUN = blood urea nitrogen. LT = low temperature; NT = normal temperature; NE = normal energy; HE = high energy; LF = low fat; HF = high fat.

**Table 10 animals-14-03061-t010:** Top 20 major differential metabolites of Songliao black fattening pigs.

NO.	Name	*p* Value	*m*/*z*	rt (s)
1	*cis*-4,7,10,13,16,19-docosahexaenoic acid	0.000056632	327.23299	46.4089
2	Phenaceturic acid	0.000100300	192.0665	210.0405
3	Enalapril	0.000482704	375.18477	204.71
4	Deoxyinosine	0.000495956	251.09591	29.8413
5	dl-Serine	0.000887771	104.03526	412.376
6	2-Isopropylmalic acid	0.001012028	197.04298	73.38165
7	Methyl hexadecanoate	0.001683752	315.25428	65.44865
8	Lignoceric acid	0.002444621	367.35804	54.9367
9	1-naphthol	0.002802418	143.07138	38.08195
10	PC (16:0/16:0)	0.003197535	732.55094	179.0265
11	Caffeic acid	0.003835029	179.05613	418.742
12	Behenic acid	0.005607347	339.32659	48.37855
13	Phenylacetic acid	0.006870292	135.02013	422.991
14	Terephthalic acid	0.008143112	165.04141	447.009
15	Shikonin	0.008969588	287.08063	36.0595
16	1-(3-pyridyl)-1-butanone-4-carboxylic acid	0.011053635	178.0509	215.957
17	3-methyl-2-oxopentanoate	0.01257369	129.05567	60.8506
18	Ile-Pro	0.012863312	227.06746	115.157
19	Glutamic acid	0.012985606	146.04591	416.0745
20	Octadecanoic acid	0.013591749	283.26423	48.2601

## Data Availability

The original contributions presented in the study are included in the article; further inquiries can be directed to the corresponding authors.

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
