# Peer review of "Effects of Dietary Energy Profiles on Energy Metabolic Partition and Excreta in Songliao Black Pigs Under Different Ambient Temperature"

_animals, 2024, doi:10.3390/ani14213061_

Round 1

Reviewer 1 Report

Comments and Suggestions for Authors

This study aimed to evaluate the performance of fattening pigs under different energy levels, temperature environments, and feed energy sources. Overall the manuscript is well-written. Some issues need to be addressed by the authors before the manuscript can be published. When the similarity index file was checked, it seemed that 28% is acceptable but it can be reduced to be lower than 25%, I encourage the authors to work on or rewrite the result section to reduce the similarity  

-        As per the style of the journal, please include the SIMPLE ABSTRACT before the abstract.

Line 10: what does N stands for? Do authors mean normal?

Line 13: Please run a power test analysis to validate whether 4 pigs per treatment is enough

Line 13: more information is needed about the housing system and whether the pigs were housed individually or in groups

Line 14: is a 6-day trial enough to draw a good conclusion about the treatments?

Line 15: here and elsewhere in the manuscript, please delete the word “significant” since the p-value is provided.

Line 16: although “OXPRO or OXFAT” is well-known, please explain it to the readers

Lines 17-18: replace “The energy digestibility ……were significantly higher” with “The digestibility of energy, protein, and fat were higher”

Line 20: Please define “RQ”

Line 23: write BUN and GLU in full the first time they appear

Lines 26-27: are the mentioned information results or conclusions?

Line 29: Do any results reflect the reduction in harmful gas emissions?

Line 29-30: please rewrite

Line 30: Please provide the reader with a clear and concise conclusion to reflect the results.  

Line 35-44: Please give references to support this information

Line 55: Please add “,” after “temperature”

Line 62-69: Please give references to support this information

Line 86: what was the age of the pigs at the beginning of the study?

Line 91: replace “the” with “The”

Line 104: Why is the feed fixed?

Line 189: the section of the results is well-written and well-detailed.

Line 348-350: no need to mention the name of the breed three times, please keep only one

Line 357: he ????? please clarify

Lines 389-394: these are results, no need to mention them again and focus on explaining these results

Lines 341-423: this part of the manuscript should include only the discussion not having the results as well, please omit the results in this section

Line 419, 438: here and elsewhere, please don’t start any statement with an abbreviation

Line 486: well-written

Author Response

Please see the attachment “animals-3263750” for details of the changes.

Comments 1: This study aimed to evaluate the performance of fattening pigs under different energy levels, temperature environments, and feed energy sources. Overall the manuscript is well-written. Some issues need to be addressed by the authors before the manuscript can be published. When the similarity index file was checked, it seemed that 28% is acceptable but it can be reduced to be lower than 25%, I encourage the authors to work on or rewrite the result section to reduce the similarity.

Response 1: Thank you for pointing out the problem. According to your suggestions, we have made changes in the manuscript to reduce the duplication rate.

Comments 2: As per the style of the journal, please include the SIMPLE ABSTRACT before the abstract.

Response 2: Thanks a lot for your suggestion. We have added a SIMPLE ABSTRACT before the abstract. (Line 7-12)

Comments 3: Line 10: what does N stands for? Do authors mean normal?

Response 3: Thank you for pointing out the problem. N stands for normal and has been added to explain. (Line 16)

Comments 4: Line 13: Please run a power test analysis to validate whether 4 pigs per treatment is enough.

Response 4: Thanks a lot for your suggestions. Because we use precision metabolic devices that are expensive to run and are limited in number, the results of the trials are relatively reproducible, we believe that four pigs per replicate is reasonable.

Comments 5: Line 13: more information is needed about the housing system and whether the pigs were housed individually or in groups.

Response 5: Thanks a lot for your suggestions. We have added the sentence “One pig per respiratory metabolic chamber in a single cage.” (Line 21)

Comments 6: Line 14: is a 6-day trial enough to draw a good conclusion about the treatments?

Response 6: Thanks a lot for your suggestion. We refer to the guidance in the textbook (Yang F. Animal nutrition [M]. China Agricultural Press, 2004.) that the formal test period for metabolic testing is typically 5-7 days.

Comments 7: Line 15: here and elsewhere in the manuscript, please delete the word “significant” since the p-value is provided.

Response 7: Thank you for pointing out the problem. We have deleted  “significant” and “significantly”in the full text.

Comments 8: Line 16: although “OXPRO or OXFAT” is well-known, please explain it to the readers.

Response 8: Thank you for pointing out the problem. We have already explained OXPRO and OXFAT. (Line 23)

Comments 9: Lines 17-18: replace “The energy digestibility ……were significantly higher” with “The digestibility of energy, protein, and fat were higher”.

Response 9: Thank you for pointing out the problem. We have deleted "significantly". (Line 25) 

Comments 10: Line 20: Please define “RQ”.

Response 10: Thank you for pointing out the problem. We have defined “RQ”. (Line 26)

Comments 11: Line 23: write BUN and GLU in full the first time they appear.

Response 11: Thank you for pointing out the problem. We have added the full names of BUN and GLU. (Line 29)

Comments 12: Lines 26-27: are the mentioned information results or conclusions?

Response 12: Thanks a lot for your suggestions. This sentence is a result of the non-targeted metabolomics section.

Comments 13: Line 29: Do any results reflect the reduction in harmful gas emissions?

Response 13: Thank you for pointing out the problem. We apologize for the lack of accuracy in the description in the original manuscript. It has been changed to greenhouse gases, and our research demonstrates a reduction in the greenhouse gas CO2 . (Line 32)

Comments 14: Line 29-30: please rewrite.

Response 14: Thank you for pointing out the problem. We have rewritten the conclusion section. (Line 30-33)

Comments 15: Line 30: Please provide the reader with a clear and concise conclusion to reflect the results.

Response 15: Thank you for pointing out the problem. We have rewritten the conclusion section. (Line 30-33)

Comments 16: Line 35-44: Please give references to support this information

Response 16: Thanks a lot for your suggestions. We have provided references to support this information.(Line 38-47)

Comments 17: Line 55: Please add “,” after “temperature”

Response 17: Thank you for pointing out the problem. We have added “,” after “temperature”. (Line 58)

Comments 18: Line 62-69: Please give references to support this information

Response 18: Thanks a lot for your suggestions. We have provided references to support this information. (Line 70)

Comments 19: Line 86: what was the age of the pigs at the beginning of the study?

Response 19: Thank you for pointing out the problem. the age of the pigs are 170 days, we have added to the manuscript. (Line 96)

Comments 20: replace “the” with “The”

Response 20: Thank you for pointing out the problem. We have replaced “the” with “The”. (Line 94)

Comments 21: Line 104: Why is the feed fixed?

Response 21: Thank you for your question. Because fixed feed intake is more convenient for us to count the differences between growth performance and metabolic indexes of pigs, if free-feeding, it may not reflect the results of differences we want in some indexes. (Line 108)

Comments 22: Line 189: the section of the results is well-written and well-detailed.

Response 22: We are honored for your recognition.

Comments 23: Line 348-350: no need to mention the name of the breed three times, please keep only one.

Response 23: Thank you for pointing out the problem. We have rewritten the passage. (Line 349-351)

Comments 24: Line 357: he ????? please clarify

Response 24: Thank you for pointing out the problem. Due to an error on our part, we did not capitalize “he”, which has now been corrected. (Line 357)

Comments 25: Lines 389-394: these are results, no need to mention them again and focus on explaining these results.

Response 25: Thank you for pointing out the problem. We have deleted the unnecessary results section.

Comments 26: Lines 341-423: this part of the manuscript should include only the discussion not having the results as well, please omit the results in this section

Response 26: Thank you for pointing out the problem. We have deleted the unnecessary results section.

Comments 27: Line 419, 438: here and elsewhere, please don’t start any statement with an abbreviation

Response 27: Thank you for pointing out the problem. We have replaced the abbreviation at the beginning of the sentence. (Line 412,430,436,438)

Comments 28: Line 486: well-written  

Response 28: Thanks again for the affirmation, we really appreciate it!

Reviewer 2 Report

Comments and Suggestions for Authors

This manuscript investigates the effects of different dietary energy levels and sources on the energy metabolism, nutrient digestibility, and growth performance of Songliao black fattening pigs under two temperature conditions (10°C and 20°C). The study uses a 2x2x2 factorial design, considering two feed energy levels (14.02 MJ/kg and 15.14 MJ/kg) and two energy sources (low-fat vs. high-fat diets). The authors aim to provide insights into optimizing energy utilization efficiency and growth performance in cold environments, emphasizing the potential benefits of high-energy and high-fat diets in enhancing pig growth and reducing environmental impact.

Line 11: the meanings of the acronyms LF (low-fat) and HF (high-fat) are provided, but the acronyms LT and NT (referring to low temperature and normal temperature) are used earlier without defining them. It is important to define all acronyms upon first use for clarity. Additionally, in the abstract, acronyms should be avoided altogether. Instead, refer directly to the groups as the low-fat group, high-fat group, low temperature, and normal temperature, to ensure that the abstract remains clear and understandable for all readers.

Line 23: the same issue with acronyms occurs. Acronyms like RQ, OXPRO, OXFAT, BUN and GLU are used without prior explanation and should not be used in the abstract.

Line 88: acronyms such as LT and NT are used for the first time in the text, so they should be fully spelled out before using the acronyms.

TABLE 1: the notation (60-100 kg) should be clarified to ensure it is clear what this range refers to.

All acronyms such as NELF, NEHF, HELF, and HEHF should be fully explained in the table footer to ensure clarity for the readers.

The footer of Table 1 contains acronyms which need to be explained in full.

TABLE 2, 3, 4, 5, 6, 7, 8, 9: All acronyms should be fully explained in the table footer to ensure clarity for the readers.

Figure 1: It appears that Figure 1 is missing from the manuscript.

SIMPLE SUMMARY

The Simple Summary is missing and should be written at the beginning of the manuscript. This section is necessary to provide a brief, non-technical overview of the study for a broader audience. Including this summary will improve the accessibility of the paper and align with the journal’s requirements.

ABSTRACT

While the key results are presented, the abstract lacks broader implications of the findings, such as how they contribute to precision livestock nutrition.

Acronyms should be avoided, and the full terms should be used instead. This will ensure clarity for readers who may not be familiar with the specific abbreviations, particularly in the abstract, where a clear and concise overview of the study is essential.

INTRODUCTION

The introduction successfully sets up the context of the study, highlighting the importance of optimizing energy metabolism in pigs, particularly under different environmental conditions. It could be improved by providing a more specific hypothesis or research question early in the text. While the topic is relevant, there is room for a more focused research objective, which would guide the reader toward the study's unique contribution.

MATERIALS AND METHODS

The description of the experimental design (2x2x2 factorial array), animal handling, and data collection methods is comprehensive. The use of indirect calorimetry, biochemical assays, and metabolomic analysis is well-detailed, giving the study methodological rigor.

There is no mention of whether the assumptions of ANOVA (normality, homogeneity of variance) were tested.

RESULTS

The results are organized in a logical manner, progressing from general growth performance to more specific analyses of energy metabolism, nutrient digestibility, and biochemical markers. The use of tables helps to clearly present the data.

DISCUSSION

The discussion tends to be too descriptive and lacks a strong critical analysis of the study’s limitations and implications. For instance, while the benefits of high-fat diets in cold environments are discussed, the potential downsides (e.g., cost, animal health, environmental impact) are not explored. The discussion would be improved by addressing the limitations of the study, such as the small sample size, short duration of the experiment, and whether the results can be generalized to other pig breeds or environmental conditions. Furthermore, the implications for practical applications in animal nutrition or recommendations for future research should be expanded upon.

CONCLUSION

The conclusion succinctly summarizes the key findings, reinforcing the idea that high-fat and high-energy diets improve energy metabolism and growth in cold environments.

Author Response

Please refer to the attachment “animals-3263750(2)” for changes in the manuscript.

Comments 1: Line 11: the meanings of the acronyms LF (low-fat) and HF (high-fat) are provided, but the acronyms LT and NT (referring to low temperature and normal temperature) are used earlier without defining them. It is important to define all acronyms upon first use for clarity. Additionally, in the abstract, acronyms should be avoided altogether. Instead, refer directly to the groups as the low-fat group, high-fat group, low temperature, and normal temperature, to ensure that the abstract remains clear and understandable for all readers.

Response 1: Thank you for pointing out the problem. We have replaced all the acronyms in the abstract. (Line 13-35)

Comments 2: Line 23: the same issue with acronyms occurs. Acronyms like RQ, OXPRO, OXFAT, BUN and GLU are used without prior explanation and should not be used in the abstract.

Response 2: Thank you for pointing out the problem. Same as the previous question, We have replaced all the acronyms in the abstract. (Line 13-35)

Comments 3: Line 88: acronyms such as LT and NT are used for the first time in the text, so they should be fully spelled out before using the acronyms.

Response 3: Thank you for pointing out the problem. We have fully spelled out the acronyms that appear for the first time in the full text. (Line 95, 96, 203)

Comments 4: TABLE 1: the notation (60-100 kg) should be clarified to ensure it is clear what this range refers to.

Response 4: Thank you for pointing out the problem. This is a weight range based on the recommendations of the local breed Songliao Black Pig.

Comments 5: All acronyms such as NELF, NEHF, HELF, and HEHF should be fully explained in the table footer to ensure clarity for the readers. The footer of Table 1 contains acronyms which need to be explained in full.

Response 5: Thank you for pointing out the problem. We have added instructions to the table footer. (Line 118-120)

Comments 6: TABLE 2, 3, 4, 5, 6, 7, 8, 9: All acronyms should be fully explained in the table footer to ensure clarity for the readers.

Response 6: Thank you for pointing out the problem. We have added instructions to the table footer. (Line 210, 249, 256, 279, 294, 310, 331, 340)

Comments 7: Figure 1: It appears that Figure 1 is missing from the manuscript.

Response 7: Thank you for pointing out the problem. Probably because of the formatting, we have added Figure 1. (Line 373)

Comments 8: SIMPLE SUMMARY The Simple Summary is missing and should be written at the beginning of the manuscript. This section is necessary to provide a brief, non-technical overview of the study for a broader audience. Including this summary will improve the accessibility of the paper and align with the journal’s requirements.

Response 8: Thank you for pointing out the problem. We have added a SIMPLE SUMMARY before the abstract. (Line 7-12)

Comments 9: ABSTRACT While the key results are presented, the abstract lacks broader implications of the findings, such as how they contribute to precision livestock nutrition.

Response 9: Thanks a lot for your suggestion. We have made some changes to the abstract. (Line 32-35)

Comments 10: INTRODUCTION The introduction successfully sets up the context of the study, highlighting the importance of optimizing energy metabolism in pigs, particularly under different environmental conditions. It could be improved by providing a more specific hypothesis or research question early in the text. While the topic is relevant, there is room for a more focused research objective, which would guide the reader toward the study's unique contribution.

Response 10: Thanks a lot for your suggestion. We have partially revised the introduction to identify more focused research objective. (Line 46-51)

Comments 11: MATERIALS AND METHODS There is no mention of whether the assumptions of ANOVA (normality, homogeneity of variance) were tested.

Response 11: Thank you for pointing out the problem. Since the treatments were 4 pigs per replicate and the data were relatively well parallelized, we regarded them as variance aligned based on the general understanding that.

Comments 12: DISCUSSION The discussion tends to be too descriptive and lacks a strong critical analysis of the study’s limitations and implications. For instance, while the benefits of high-fat diets in cold environments are discussed, the potential downsides (e.g., cost, animal health, environmental impact) are not explored. The discussion would be improved by addressing the limitations of the study, such as the small sample size, short duration of the experiment, and whether the results can be generalized to other pig breeds or environmental conditions. Furthermore, the implications for practical applications in animal nutrition or recommendations for future research should be expanded upon.

Response 12: Thanks a lot for your suggestion. Based on your suggestions, we have modified the discussion section. (Line 383, 422, 471, 500, 522)

Round 2

Reviewer 1 Report

Comments and Suggestions for Authors

The authors did a great job improving the manuscript's presentation and quality.

Just a few editorials need to be addressed

Lines 25-26: compare to what?

lines 147, 157, 405: replace "gross energy (GE)" with "GE"

line 204: replace "Feed to Gain Ratio" with "feed to gain ratio"

lines 405-407: please keep the abbreviations since they were defined earlier     

Reviewer 2 Report

Comments and Suggestions for Authors

Dear authors,

I would like to express my sincere gratitude for your prompt and thorough attention to all the revisions and suggestions I provided. Your dedication and meticulous work have significantly enhanced the quality of the manuscript, demonstrating your commitment to scientific and academic excellence.

I greatly appreciate your willingness to implement the necessary changes and to address each of the points raised with rigor, which reflects your professionalism and collaborative spirit. I am confident that these adjustments will enrich the presented results and facilitate the understanding of the content by the scientific community.